# DeeperGCN: Training Deeper GCNs with Generalized Aggregation Functions

## Abstract

Graph Convolutional Networks (GCNs) have been drawing significant attention with the power of representation learning on graphs. Recent works developed frameworks to train deep GCNs. Such works show impressive results in tasks like point cloud classification and segmentation, and protein interaction prediction. In this work, we study the performance of such deep models in large scale graph datasets from the Open Graph Benchmark (OGB). In particular, we look at the effect of adequately choosing an aggregation function, and its effect on final performance. Common choices of aggregation are *mean*, *max*, and *sum*. It has shown that GCNs are sensitive to such aggregations when applied to different datasets. We further validate this point and propose to alleviate it by introducing a novel *Generalized Aggregation Function*. Our new aggregation not only covers all commonly used ones, but also can be tuned to learn customized functions for different tasks. Our generalized aggregation is fully differentiable, and thus its parameters can be learned in an end-to-end fashion. We add our generalized aggregation into a deep GCN framework and show it achieves state-of-the-art results in six benchmarks from OGB.

## 1 Introduction

The rise of availability of non-Euclidean data (Bronstein et al., 2017) has recently shed interest into the topic of Graph Convolutional Networks (GCNs). GCNs provide powerful deep learning architectures for irregular data, like point clouds and graphs. GCNs have proven valuable for applications in social networks (Tang & Liu, 2009), drug discovery (Zitnik & Leskovec, 2017; Wale et al., 2008), recommendation engines (Monti et al., 2017b; Ying et al., 2018), and point clouds (Wang et al., 2018; Li et al., 2019b). Recent works looked at frameworks to train deeper GCN architectures (Li et al., 2019b;a). These works demonstrate how increased depth leads to state-of-the-art performance on tasks like point cloud classification and segmentation, and protein interaction prediction. The power of deep models become more evident with the introduction of more challenging and large-scale graph datasets. Such datasets were recently introduced in the Open Graph Benchmark (OGB) (Hu et al., 2020), for tasks of *node classification*, *link prediction*, and *graph classification*.

Graph convolutions in GCNs are based on the notion of message passing (Gilmer et al., 2017). To compute a new node feature at each GCN layer, information is aggregated from the node and its connected neighbors. Given the nature of graphs, aggregation functions must be permutation invariant. This property guarantees invariance/equivariance to isomorphic graphs (Battaglia et al., 2018; Xu et al., 2019b; Maron et al., 2019a). Popular choices for aggregation functions are *mean* (Kipf & Welling, 2016), *max* (Hamilton et al., 2017), and *sum* (Xu et al., 2019b). Recent works suggest different aggregations have different performance impact depending on the task. For example, *mean* and *sum* perform best in node classification (Kipf & Welling, 2016), while *max* is favorable for dealing with 3D point clouds (Qi et al., 2017; Wang et al., 2019). Currently, all works rely on empirical analysis to choose aggregation functions.

In DeepGCNs (Li et al. (2019b)), the authors complement aggregation functions with residual and dense connections, and dilated convolutions, in order to train very deep GCNs. Equipped with these new modules, GCNs with more than 100 layers can be reliably trained. Despite the potential of these new modules (Kipf & Welling, 2016; Hamilton et al., 2017; Veličković et al., 2018; Xu et al., 2019a), it is still unclear if they are the ideal choice for DeepGCNs when handling large-scale graphs.

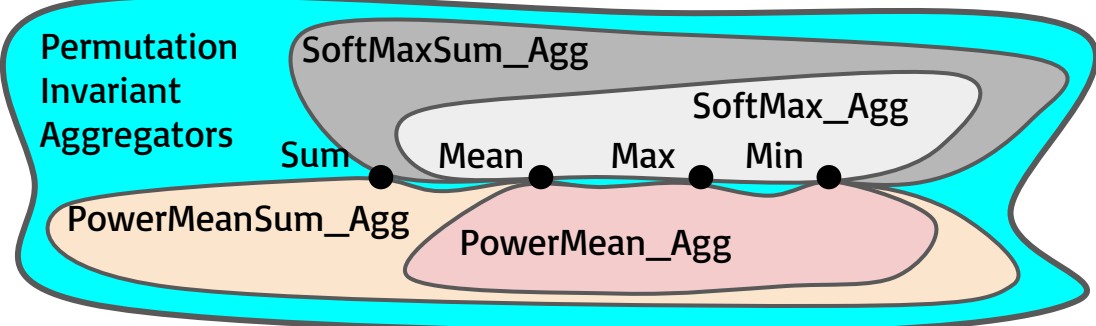

Figure 1: Illustration of Generalized Message Aggregation Functions

In this work, we analyze the performance of GCNs on large-scale graphs. In particular, we look at the effect of aggregation functions in performance. We unify aggregation functions by proposing a novel *Generalized Aggregation Function* (Figure 1) suited for graph convolutions. We show how our function covers all commonly used aggregations (*mean*, *max*, and *sum*), and its parameters can be tuned to learn customized functions for different tasks. Our novel aggregation is fully differentiable and can be learned in an end-to-end fashion in a deep GCN framework. In our experiments, we show the performance of baseline aggregations in various large-scale graph datasets. We then introduce our generalized aggregation and observe improved performance with the correct choice of aggregation parameters. Finally, we demonstrate how learning the parameters of our generalized aggregation, in an end-to-end fashion, leads to state-of-the-art performance in several OGB benchmarks. Our analysis indicates the choice of suitable aggregations is imperative to the performance of different tasks. A differentiable generalized aggregation function ensures the correct aggregation is used for each learning scenario.

We summarize our contributions as two-fold: **(1)** We propose a novel *Generalized Aggregation Function*. This new function is suitable for GCNs, as it enjoys a permutation invariant property. We show how our generalized aggregation covers commonly used functions such as *mean*, *max*, and *sum* in graph convolutions. Additionally, we show how its parameters can be tuned to improve performance on diverse GCN tasks. Since this new function is fully differentiable, we show how its parameters can be learned in an end-to-end fashion. **(2)** We run extensive experiments on seven datasets from the Open Graph Benchmark (OGB). Our results show that combining depth with our generalized aggregation function achieves state-of-the-art in several of these benchmarks.

## 2 RELATED WORK

**Graph Convolutional Networks (GCNs).** Current GCN algorithms can be divided into two categories: spectral-based and spatial-based. Based on spectral graph theory, Bruna et al. (2013) firstly developed graph convolutions using the Fourier basis of a given graph in the spectral domain. Later, many methods proposed to apply improvements, extensions, and approximations on spectral-based GCNs (Kipf & Welling, 2016; Defferrard et al., 2016; Henaff et al., 2015; Levie et al., 2018; Li et al., 2018; Wu et al., 2019). Spatial-based GCNs (Scarselli et al., 2008; Hamilton et al., 2017; Monti et al., 2017a; Niepert et al., 2016; Gao et al., 2018; Xu et al., 2019b; Veličković et al., 2018) define graph convolution operations directly on the graph by aggregating information from neighbor nodes. To address the scalability issue of GCNs on large-scale graphs, two main categories of algorithms exist: sampling-based (Hamilton et al., 2017; Chen et al., 2018b; Li et al., 2018; Chen et al., 2018a; Zeng et al., 2020) and clustering-based (Chiang et al., 2019).

**Training Deep GCNs.** Despite the rapid and fruitful progress of GCNs, most prior work employs shallow GCNs. Several works attempt different ways of training deeper GCNs (Hamilton et al., 2017; Armeni et al., 2017; Rahimi et al., 2018; Xu et al., 2018). However, all these approaches are limited to 10 layers of depth, after which GCN performance would degrade because of vanishing gradient and over-smoothingLi et al. (2018). Inspired by the merits of training deep CNN-based networks (He et al., 2016a; Huang et al., 2017; Yu & Koltun, 2016), DeepGCNs (Li et al., 2019b) propose to train very deep GCNs (56 layers) by adapting residual/dense connections

(ResGCN/DenseGCN) and dilated convolutions to GCNs. DeepGCN variants achieve state-of-the art results on S3DIS point cloud semantic segmentation (Armeni et al., 2017) and the PPI dataset. Many recent works focus on further addressing this phenomenon (Klicpera et al., 2019; Rong et al., 2020; Zhao & Akoglu, 2020; Chen et al., 2020; Gong et al., 2020; Rossi et al., 2020). In particular, Klicpera et al. (2019) propose a PageRank-based message passing mechanism involving the root node in the loop. Alternatively, DropEdge (Rong et al., 2020) randomly removes edges from the graph, and PairNorm (Zhao & Akoglu, 2020) develops a novel normalization layer. We find that the choice of aggregation may also limit the power of deep GCNs. In this work, we thoroughly study the important relation between aggregation functions and deep GCN architectures.

**Aggregation Functions for GCNs.** GCNs update a node's feature vector by aggregating feature information from its neighbors in the graph. Many different neighborhood aggregation functions that possess a permutation invariant property have been proposed (Hamilton et al., 2017; Veličković et al., 2018; Xu et al., 2019b). Specifically, Hamilton et al. (2017) examine mean, max, and LSTM aggregators, and they empirically find that max and LSTM achieve the best performance. Graph attention networks (GATs) (Veličković et al., 2018) employ the attention mechanism (Bahdanau et al., 2015) to obtain different and trainable weights for neighbor nodes by learning the attention between their feature vectors and that of the central node. Thus, the aggregator in GATs operates like a learnable weighted mean. Furthermore, Xu et al. (2019b) propose a GCN architecture, denoted Graph Isomorphism Network (GIN), with a sum aggregation that has been shown to have high discriminative power according to the Weisfeiler-Lehman (WL) graph isomorphism test (Weisfeiler & Lehman, 1968). In this work, we propose generalized message aggregation functions, a new family of aggregation functions, that generalizes conventional aggregators including *mean*, *max* and *sum*. With the nature of differentiablity and continuity, generalized message aggregation functions provide a new perspective for designing GCN architectures.

## 3 REPRESENTATION LEARNING ON GRAPHS

**Graph Representation.** A graph $\mathcal{G}$ is usually defined as a tuple of two sets $\mathcal{G} = (\mathcal{V}, \mathcal{E})$, where $\mathcal{V} = \{ v_1, v_2, ..., v_N \}$ and $\mathcal{E} \subseteq \mathcal{V} \times \mathcal{V}$ are the sets of vertices and edges, respectively. If an edge $e_{ij} = (v_i, v_j) \in \mathcal{E}$ for an undirected graph, $e_{ij}$ is an edge connecting vertices $v_i$ and $v_j$; for a directed graph, $e_{ij}$ is an edge directed from $v_i$ to $v_j$. Usually, a vertex $v$ and an edge $e$ in the graph are associated with vertex features $\mathbf{h}_v \in \mathbb{R}^D$ and edge features $\mathbf{h}_e \in \mathbb{R}^C$ respectively.[1]

**GCNs for Learning Graph Representation.** We define a general graph representation learning operator $\mathcal{F}$, which takes as input a graph $\mathcal{G}$ and outputs a transformed graph $\mathcal{G}'$, *i.e.* $\mathcal{G}' = \mathcal{F}(\mathcal{G})$. The features or even the topology of the graph can be learned or updated after the transformation $\mathcal{F}$. Typical graph representation learning operators usually learn latent features or representations for graphs such as DeepWalk (Perozzi et al., 2014), Planetoid (Yang et al., 2016), Node2Vec (Grover & Leskovec, 2016), Chebyshev graph CNN (Defferrard et al., 2016), GCN (Kipf & Welling, 2016), Neural Message Passing Network (MPNN) (Gilmer et al., 2017), GraphSage (Hamilton et al., 2017), GAT (Veličković et al., 2018) and GIN (Xu et al., 2019b). In this work, we focus on the GCN family and its message passing framework (Gilmer et al., 2017; Battaglia et al., 2018). To be specific, message passing based on the GCN operator $\mathcal{F}$ operating on vertex $v \in \mathcal{V}$ at the $l$-th layer is defined as follows:

$$\mathbf{m}_{vu}^{(l)} = \boldsymbol{\rho}^{(l)}(\mathbf{h}_v^{(l)}, \mathbf{h}_u^{(l)}, \mathbf{h}_{e_{vu}}^{(l)}), \ \forall u \in \mathcal{N}(v) \tag{1}$$

$$\mathbf{m}_v^{(l)} = \boldsymbol{\zeta}^{(l)}(\{ \mathbf{m}_{vu}^{(l)} \mid u \in \mathcal{N}(v) \}) \tag{2}$$

$$\mathbf{h}_v^{(l+1)} = \boldsymbol{\phi}^{(l)}(\mathbf{h}_v^{(l)}, \mathbf{m}_v^{(l)}), \tag{3}$$

where $\boldsymbol{\rho}^{(l)}, \boldsymbol{\zeta}^{(l)}$, and $\boldsymbol{\phi}^{(l)}$ are all learnable or differentiable functions for *message construction*, *message aggregation*, and *vertex update* at the $l$-th layer, respectively. For simplicity, we only consider the case where vertex features are updated at each layer. It is straightforward to extend it to edge features. Message construction function $\boldsymbol{\rho}^{(l)}$ is applied to vertex features $\mathbf{h}_v^{(l)}$ of $v$, its neighbor's features $\mathbf{h}_u^{(l)}$, and the corresponding edge features $\mathbf{h}_{e_{vu}}^{(l)}$ to construct an individual message $\mathbf{m}_{vu}^{(l)}$ for

---

[1]In some cases, vertex features or edge features are absent.

each neighbor $u \in \mathcal{N}(v)$. Message aggregation function $\boldsymbol{\zeta}^{(l)}$ is commonly a permutation invariant set function that takes as input a countable unordered message set $\{\, \mathbf{m}_{vu}^{(l)} \mid u \in \mathcal{N}(v) \,\}$, where $\mathbf{m}_{vu}^{(l)} \in \mathbb{R}^D$, and outputs a reduced or aggregated message $\mathbf{m}_v^{(l)} \in \mathbb{R}^D$. The permutation invariance of $\boldsymbol{\zeta}^{(l)}$ guarantees the invariance/equivariance to isomorphic graphs (Battaglia et al., 2018). $\boldsymbol{\zeta}^{(l)}$ can simply be a symmetric function such as *mean* (Kipf & Welling, 2016), *max* (Hamilton et al., 2017), or *sum* (Xu et al., 2019b). Vertex update function $\boldsymbol{\phi}^{(l)}$ combines the original vertex features $\mathbf{h}_v^{(l)}$ and the aggregated message $\mathbf{m}_v^{(l)}$ to obtain the transformed vertex features $\mathbf{h}_v^{(l+1)}$.

# 4   Beyond Mean, Max, and Sum Aggregation functions

**Property 1** (Graph Isomorphic Equivariance)**.** *If a message aggregation function $\boldsymbol{\zeta}$ is permutation invariant to the message set $\{\, \mathbf{m}_{vu} \mid u \in \mathcal{N}(v) \,\}$, then the message passing based GCN operator $\mathcal{F}$ is equivariant to graph isomorphism,* i.e. *for any isomorphic graphs $\mathcal{G}_1$ and $\mathcal{G}_2 = \sigma \star \mathcal{G}_1$, $\mathcal{F}(\mathcal{G}_2) = \sigma \star \mathcal{F}(\mathcal{G}_1)$, where $\star$ denotes a permutation operator on graphs.*

The invariance and equivariance properties on sets or GCNs have been discussed in many recent works. Zaheer et al. (2017) propose DeepSets based on permutation invariance and equivariance to deal with sets as inputs. Maron et al. (2019c) show the universality of invariant GCNs to any continuous invariant function. Keriven & Peyré (2019) further extend it to the equivariant case. Maron et al. (2019b) compose networks by proposing invariant or equivariant linear layers and show that their models are as powerful as any MPNN (Gilmer et al., 2017). In this work, we study permutation invariant functions of GCNs, which enjoy these proven properties.

## 4.1   Generalized Message Aggregation Functions

To embrace the properties of invariance and equivariance (Property 1), many works in the graph learning field tend to use simple permutation invariant functions like *mean* (Kipf & Welling, 2016), *max* (Hamilton et al., 2017) and *sum* (Xu et al., 2019b). Inspired by the Weisfeiler-Lehman (WL) graph isomorphism test (Weisfeiler & Lehman, 1968), Xu et al. (2019b) propose a theoretical framework and analyze the representational power of GCNs with *mean*, *max* and *sum* aggregators. Although *mean* and *max* aggregators are proven to be less powerful than *sum* according to the WL test in (Xu et al., 2019b), they are found to be quite effective in the tasks of node classification (Kipf & Welling, 2016; Hamilton et al., 2017) and 3D point cloud processing (Qi et al., 2017; Wang et al., 2019) To go beyond these simple aggregation functions and study their characteristics, we define generalized aggregation functions in the following.

**Definition 2** (Generalized Message Aggregation Functions)**.** We define a generalized message aggregation function $\boldsymbol{\zeta_z}(\cdot)$ as a function that is parameterized by a continuous variable $\boldsymbol{z}$ to produce a family of permutation invariant set functions, *i.e.* $\forall \boldsymbol{z}$, $\boldsymbol{\zeta_z}(\cdot)$ is permutation invariant to the order of messages in the set $\{\, \mathbf{m}_{vu} \mid u \in \mathcal{N}(v) \,\}$.

In order to subsume the popular *mean* and *max* aggregations into the generalized space, we further define *generalized mean-max aggregation* parameterized by a scalar for message aggregation.

**Definition 3** (Generalized Mean-Max Aggregation)**.** If there exists a pair of $x$ say $x_1$, $x_2$ such that for any message set $\lim_{x \to x_1} \boldsymbol{\zeta}_x(\cdot) = \mathrm{Mean}(\cdot)$ [2] and $\lim_{x \to x_2} \boldsymbol{\zeta}_x(\cdot) = \mathrm{Max}(\cdot)$, then $\boldsymbol{\zeta}_x(\cdot)$ is a generalized mean-max aggregation function.

The nice properties of generalized mean-max aggregation functions can be summarized as follows: **(1)** they provide a large family of permutation invariant aggregation functions; **(2)** they are continuous and differentiable in $x$ and are potentially learnable; **(3)** it is possible to interpolate between $x_1$ and $x_2$ to find a better aggregator than *mean* and *max* for a given task. To empirically validate these properties, we propose two families of generalized mean-max aggregation functions based on Definition 3, namely *SoftMax aggregation* and *PowerMean aggregation*.

**Proposition 4** (SoftMax Aggregation)**.** *Given any message set $\{\, \mathbf{m}_{vu} \mid u \in \mathcal{N}(v) \,\}$, $\mathbf{m}_{vu} \in \mathbb{R}^D$, SoftMax_Agg$_\beta(\cdot)$ is a generalized mean-max aggregation function, where SoftMax_Agg$_\beta(\cdot)$ =*

---

[2]Mean$(\cdot)$ denotes the arithmetic mean.

$\sum_{u \in \mathcal{N}(v)} \frac{\exp(\beta \mathbf{m}_{vu})}{\sum_{i \in \mathcal{N}(v)} \exp(\beta \mathbf{m}_{vi})} \cdot \mathbf{m}_{vu}$. *Here, $\beta$ is a continuous variable called an inverse temperature.*

The SoftMax function with a temperature has been studied in many machine learning areas, *e.g.* Energy-Based Learning (LeCun et al., 2006), Knowledge Distillation (Hinton et al., 2015) and Reinforcement Learning (Gao & Pavel, 2017). Here, for low inverse temperatures $\beta$, SoftMax_Agg$_\beta(\cdot)$ behaves like a mean aggregation. For high inverse temperatures, it approaches a max aggregation. Formally, $\lim_{\beta \to 0}$SoftMax_Agg$_\beta(\cdot) = $ Mean$(\cdot)$ and $\lim_{\beta \to \infty}$SoftMax_Agg$_\beta(\cdot) = $ Max$(\cdot)$. It can be regarded as a weighted summation that depends on the inverse temperature $\beta$ and the values of the elements themselves. The full proof of Proposition 4 is in the Appendix.

**Proposition 5** (PowerMean Aggregation). *Given any message set $\{\mathbf{m}_{vu} \mid u \in \mathcal{N}(v)\}$, $\mathbf{m}_{vu} \in \mathbb{R}_+^D$, PowerMean_Agg$_p(\cdot)$ is a generalized mean-max aggregation function, where PowerMean_Agg$_p(\cdot) = (\frac{1}{|\mathcal{N}(v)|} \sum_{u \in \mathcal{N}(v)} \mathbf{m}_{vu}^p)^{1/p}$. Here, $p$ is a non-zero, continuous variable denoting the $p$-th power.*

Quasi-arithmetic mean (Kolmogorov & Castelnuovo, 1930) was proposed to unify the family of mean functions. Power mean is one member of the Quasi-arithmetic mean family. It is a generalized mean function that includes harmonic mean, geometric mean, arithmetic mean, and quadratic mean. The main difference between Proposition 4 and 5 is that Proposition 5 only holds when message features are all positive, *i.e.* $\mathbf{m}_{vu} \in \mathbb{R}_+^D$. In particular, we have PowerMean_Agg$_{p=1}(\cdot) = $ Mean$(\cdot)$ and $\lim_{p \to \infty}$PowerMean_Agg$_p(\cdot) = $ Max$(\cdot)$. PowerMean_Agg$_p(\cdot)$ becomes the harmonic or the geometric mean aggregation when $p = -1$ or $p \to 0$, respectively. See the Appendix for the proof.

To enhance expressive power according to the WL test (Xu et al., 2019b), we generalize the function space to cover the *sum* aggregator by introducing another control variable on the degree of vertices.

**Proposition 6** (Generalized Mean-Max-Sum Aggregation). *Given any generalized mean-max aggregation function $\boldsymbol{\zeta}_x(\cdot)$, we can generalize the function to cover* sum *by combining it with the degree of vertices. For instance, by introducing a variable $y$, we can compose a generalized mean-max-sum aggregation function as $|\mathcal{N}(v)|^y \cdot \boldsymbol{\zeta}_x(\cdot)$. We can observe that the function becomes a Sum aggregation when $\boldsymbol{\zeta}_x(\cdot)$ is a Mean aggregation and $y = 1$. By composing with SoftMax aggregation and PowerMean aggregation, we obtain SoftMaxSum_Agg$_{(\beta, y)}(\cdot)$ and PowerMeanSum_Agg$_{(p, y)}(\cdot)$ aggregation functions, respectively.*

## 4.2 GENERALIZED AGGREGATION NETWORKS (GEN)

**Generalized Message Passing Layer.** Based on the Propositions above, we construct a simple message passing based GCN network that satisfies the conditions in Proposition 4 and 5. The key idea is to keep all the message features to be positive, so that generalized mean-max aggregation functions (SoftMax_Agg$_\beta(\cdot)$ and PowerMean_Agg$_p(\cdot)$) can be applied. We define the message construction function $\boldsymbol{\rho}^{(l)}$ as follows:

$$\mathbf{m}_{vu}^{(l)} = \boldsymbol{\rho}^{(l)}(\mathbf{h}_v^{(l)}, \mathbf{h}_u^{(l)}, \mathbf{h}_{e_{vu}}^{(l)}) = \text{ReLU}(\mathbf{h}_u^{(l)} + \mathbb{1}(\mathbf{h}_{e_{vu}}^{(l)}) \cdot \mathbf{h}_{e_{vu}}^{(l)}) + \epsilon, \ \forall u \in \mathcal{N}(v) \qquad (4)$$

where ReLU$(\cdot)$ is a rectified linear unit (Nair & Hinton, 2010) that outputs values to be greater or equal to zero, $\mathbb{1}(\cdot)$ is an indicator function being 1 when edge features exist otherwise 0, and $\epsilon$ is a small positive constant chosen to be $10^{-7}$. As the conditions are satisfied, we can choose the message aggregation function $\boldsymbol{\zeta}^{(l)}(\cdot)$ to be either SoftMax_Agg$_\beta(\cdot)$, PowerMean_Agg$_p(\cdot)$, SoftMaxSum_Agg$_{(\beta, y)}(\cdot)$, or PowerMeanSum_Agg$_{(p, y)}(\cdot)$. As for the vertex update function $\boldsymbol{\phi}^{(l)}$, we use a simple multi-layer perceptron, where $\boldsymbol{\phi}^{(l)} = \text{MLP}(\mathbf{h}_v^{(l)} + \mathbf{m}_v^{(l)})$.

**Skip Connections and Normalization.** Skip connections and normalization techniques are important to train deep GCNs. Li et al. (2019b) propose residual GCN blocks with components following the ordering: GraphConv $\to$ Normalization $\to$ ReLU $\to$ Addition. He et al. (2016b) studied the effect of ordering of ResNet components in CNNs, showing its importance. As recommended in their paper, the output range of the residual function should be $(-\infty, +\infty)$. Activation functions such as ReLU before addition may impede the representational power of deep models. Therefore, we adopt a pre-activation variant of residual connections for GCNs, which follows the ordering:

Normalization $\rightarrow$ ReLU $\rightarrow$ GraphConv $\rightarrow$ Addition. Empirically, we find that the pre-activation version performs better. In our architectures, normalization methods such as BatchNorm (Ioffe & Szegedy, 2015) or LayerNorm (Ba et al., 2016) are applied to normalize vertex features.

## 5 EXPERIMENTS

We propose *GENeralized Aggregation Networks* (GEN) equipped with generalized message aggregators. To evaluate the effectiveness of these aggregators, we perform extensive experiments on the *Open Graph Benchmark* (OGB) (Hu et al., 2020), which includes a diverse set of challenging and large-scale tasks and datasets. We first conduct a comprehensive ablation study on the task of node property prediction on *ogbn-proteins* and *ogbn-arxiv* datasets. Then, we apply our GEN framework on the node property prediction dataset (*ogbn-products*), three graph property prediction datasets (*ogbg-molhiv*, *ogbg-molpcba* and *ogbg-ppa*), and one link property prediction dataset (*ogbl-collab*).

### 5.1 EXPERIMENTAL SETUP

**Baseline Models.** The PlainGCN model stacks GCNs from 3 layers to 112 layers without skip connections. Each GCN layer uses the same message passing operator as in GEN except the aggregation function is replaced by Sum($\cdot$), Mean($\cdot$), or Max($\cdot$) aggregation. LayerNorm or BatchNorm is used in every layer before the ReLU activation function. Similar to Li et al. (2019b), we use Res-GCN layers by adding residual connections to PlainGCN following the ordering: GraphGonv $\rightarrow$ Normalization $\rightarrow$ ReLU $\rightarrow$ Addition. We construct the pre-activation version of ResGCN by changing the order of residual connections to Normalization $\rightarrow$ ReLU $\rightarrow$ GraphGonv $\rightarrow$ Addition. We denote this as ResGCN+ to differentiate it from ResGCN. The effect of residual connections can be found in Appendix A.

**ResGEN.** The ResGEN models are designed using the message passing functions described in Section 4.2. The only difference between ResGEN and ResGCN+ is that generalized message aggregators are used instead of Sum($\cdot$), Mean($\cdot$), or Max($\cdot$). For simplicity, we study generalized mean-max aggregators (*i.e.* SoftMax_Agg$_\beta(\cdot)$ and PowerMean_Agg$_p(\cdot)$) which are parameterized by only one scalar. To explore the characteristics of the generalized message aggregators, we instantiate them with different hyper-parameters. Here, we freeze the values of $\beta$ to $10^n$, where $n \in \{-3, -2, -1, 0, 1, 2, 3, 4\}$ and $p$ to $\{-1, 10^{-3}, 1, 2, 3, 4, 5, 10\}$.

**DyResGEN.** In contrast to ResGEN, DyResGEN learns variables $\beta$, $p$ or $y$ *dynamically* for every layer at every gradient descent step. By learning these variables, we avoid the need to painstakingly search for the best hyper-parameters. In doing so, DyResGEN can learn aggregation functions that adapt to the training process and the dataset. We study the potential of learning these variables for our proposed aggregators: SoftMax_Agg$_\beta(\cdot)$, PowerMean_Agg$_p(\cdot)$, SoftMaxSum_Agg$_{(\beta,y)}(\cdot)$, and PowerMeanSum_Agg$_{(p,y)}(\cdot)$.

**Datasets.** Traditional graph datasets have been shown limited and unable to provide reliable evaluation and rigorous comparison among methods (Hu et al., 2020; Dwivedi et al., 2020). Reasons include their small-scale nature, non-negligible duplication or leakage rates, unrealistic data splits, *etc*. Consequently, we conduct our experiments on the recently released datasets of Open Graph Benchmark (OGB) (Hu et al., 2020), which overcome the main drawbacks of commonly used datasets and thus are much more realistic and challenging. OGB datasets cover a variety of real-world applications and span several important domains ranging from social and information networks to biological networks, molecular graphs, and knowledge graphs. They also span a variety of prediction tasks at the level of nodes, graphs, and links/edges. In this work, experiments are performed on three OGB datasets for node property prediction, three OGB datasets for graph property prediction, and one OGB dataset for link property prediction. We introduce these seven datasets briefly in Appendix E.2. More detailed information about OGB datasets can be found in (Hu et al., 2020).

**Implementation Details.** We first perform ablation studies on the ogbn-proteins and ogbn-arxiv datasets. Then, we evaluate our model on the other datasets and compare the performances with state-of-the-art (SOTA) methods. Since the ogbn-proteins dataset is very dense and comparably large, full-batch training is infeasible when considering very deep GCNs. We simply apply a random partition to generate batches for both mini-batch training and test. We set the number of partitions to

10 for training and 5 for test, and we set the batch size to 1 subgraph. In comparison, the ogbn-arxiv dataset is relatively small, so we conduct experiments via full batch training and test in this case.

## 5.2 RESULTS

**Aggregators may Limit the Power of Deep GCNs.** Although pre-activation residual connections alleviate the effect of vanishing gradients and enable the training of deep GCNs, the choice of aggregation function is crucial to performance. In Table 1 (a) *ResGCN+*, we study how conventional aggregators (*i.e.* Sum, Mean and Max) behave on ogbn-proteins and ogbn-arxiv. We find that not all of them benefit from network depth. The aggregators perform inconsistently among different datasets and cause significant gaps in performance. For instance, the Max aggregator outperforms the other two by a large margin ($\sim 1\%$) for all network depths on ogbn-proteins, but reaches unsatisfactory results ($< 70\%$) and even becomes worse with depth increasing on ogbn-arxiv. The Mean aggregator performs the worst on ogbn-proteins, but the best (72.31%) with 28 layers on ogbn-arxiv.

Table 1: Ablation studies of aggregation functions on the ogbn-proteins and ogbn-arxiv datasets

| (a) | | | ogbn-proteins | | | | ogbn-arxiv | | |
|---|---|---|---|---|---|---|---|---|---|
| Model | #Layers | Sum | Mean | Max | SoftMax | Sum | Mean | Max | PowerMeanSum |
| ResGCN+ | 3 | 82.67 | 79.69 | **83.47** | 83.42 | 70.89 | **71.17** | 69.59 | 72.12 |
| | 7 | 83.00 | 80.84 | **84.65** | 84.81 | 71.17 | **71.83** | 69.57 | 72.31 |
| | 14 | 83.33 | 82.25 | **85.16** | 85.29 | 71.50 | **72.03** | 68.97 | 72.14 |
| | 28 | 83.98 | 83.28 | **85.26** | 85.51 | 71.32 | **72.31** | 66.91 | 72.40 |
| | 56 | 84.48 | 83.52 | **86.05** | 86.12 | – | – | – | – |
| | 112 | 85.33 | 83.40 | **85.94** | 86.15 | – | – | – | – |
| | avg. | 83.80 | 82.16 | **85.09** | 85.22 | 71.22 | **71.83** | 68.76 | 72.24 |

| (b) | ogbn-proteins | | | | SoftMax | | | |
|---|---|---|---|---|---|---|---|---|
| Model | #Layers | $10^{-3}$ | $10^{-2}$ | $10^{-1}$ | 1 | 10 | $10^2$ | $10^3$ | $10^4$ |
| ResGEN | 3 | 79.69 | 78.90 | 77.80 | 81.69 | **83.24** | 83.16 | 83.07 | 83.21 |
| | 7 | 80.81 | 80.71 | 79.83 | 83.85 | 83.98 | 84.66 | 84.60 | **84.68** |
| | 14 | 82.44 | 82.14 | 81.24 | 84.39 | **85.13** | 84.96 | 84.99 | 84.85 |
| | 28 | 83.13 | 82.47 | 81.78 | 85.08 | 85.07 | 85.35 | 85.80 | **85.82** |
| | 56 | 83.62 | 83.45 | 82.86 | 85.76 | 85.97 | **86.20** | 85.98 | 86.19 |
| | 112 | 83.50 | 83.61 | 83.16 | 85.77 | **86.38** | 86.27 | 86.27 | 86.30 |
| | avg. | 82.20 | 81.88 | 81.11 | 84.42 | 84.96 | 85.10 | 85.12 | **85.17** |

| (c) | ogbn-proteins | | | | PowerMean | | | |
|---|---|---|---|---|---|---|---|---|
| Model | #Layers | $-1$ | $10^{-3}$ | 1 | 2 | 3 | 4 | 5 | 10 |
| ResGEN | 3 | 82.34 | 81.06 | 78.52 | 80.23 | 82.01 | 81.61 | **82.89** | **82.89** |
| | 7 | 83.36 | 81.08 | 81.02 | 83.49 | 83.67 | **84.82** | 84.54 | 84.50 |
| | 14 | 83.73 | 80.64 | 82.45 | 84.15 | 84.48 | 84.64 | 85.00 | **85.08** |
| | 28 | 84.56 | 80.92 | 82.58 | 84.16 | 85.20 | **85.87** | 85.34 | 85.76 |
| | 56 | 84.46 | 80.93 | 83.49 | 85.04 | 85.68 | **85.90** | 85.64 | 85.74 |
| | 112 | 85.13 | 81.10 | 83.92 | 85.47 | 85.70 | 86.01 | 86.09 | **86.31** |
| | avg. | 83.93 | 80.95 | 82.00 | 83.76 | 84.46 | 84.81 | 84.92 | **85.05** |

| (d) | ogbn-proteins | SoftMax | | SoftMaxSum | | PowerMean | | PowerMeanSum | |
|---|---|---|---|---|---|---|---|---|---|
| Model | #Layers | Fixed | Learned | Fixed | Learned | Fixed | Learned | Fixed | Learned |
| DyResGEN | 3 | 81.69 | 83.42 | 83.06 | 83.42 | 78.52 | 82.25 | 81.70 | **83.71** |
| | 7 | 83.85 | **84.81** | 84.71 | 84.63 | 81.02 | 84.14 | 83.23 | 84.62 |
| | 14 | 84.39 | **85.29** | 84.77 | 85.03 | 82.45 | 85.04 | 83.96 | 84.83 |
| | 28 | 85.08 | 85.51 | 85.64 | 85.66 | 82.58 | 85.04 | 84.59 | **85.96** |
| | 56 | 85.76 | **86.12** | 85.63 | 85.50 | 83.49 | 85.27 | 85.37 | 85.81 |
| | 112 | 85.77 | **86.15** | 86.11 | 86.13 | 83.92 | 85.60 | 85.71 | 86.01 |
| | avg. | 84.42 | **85.22** | 84.99 | 85.06 | 82.00 | 84.56 | 84.09 | 85.16 |

**Exploring Generalized Message Aggregators.** In Table 1 (b) & (c) *ResGEN*, we examine SoftMax_Agg$_\beta(\cdot)$ and PowerMean_Agg$_p(\cdot)$ aggregators on ogbn-proteins by measuring test ROC-AUC. Since both are *generalized mean-max aggregations*, they can theoretically perform at least as good as Mean and Max through interpolation. For SoftMax_Agg, when $\beta = 10^{-3}$, it performs similarly to Mean aggregation (82.20% *vs.* 82.16%). As $\beta$ increases to $10^2$, it achieves slightly better

performance than Max aggregation. Remarkably, 112-layer ResGEN with SoftMax_Agg reaches $86.38\%$ and $86.30\%$ ROC-AUC when $\beta = 10$ and $\beta = 10^4$ respectively. For PowerMean_Agg, we find that it reaches almost the same ROC-AUC as Mean when $p = 1$ (arithmetic mean). We also observe that all other orders of mean except $p = 10^{-3}$ (akin to geometric mean) achieve better performance than the arithmetic mean. PowerMean_Agg with $p = 10$ reaches the best ROC-AUC at $86.31\%$ with 112 layers. However, due to some numerical issues in PyTorch (Paszke et al., 2019), we are not able to use larger $p$. These results empirically validate the discussion on existence of better generalized mean-max aggregators beyond mean and max in Section 4.1.

**Learning Dynamic Aggregators.** Trying out every possible aggregator or searching hyperparameters is computationally expensive. Therefore, we propose DyResGEN to explore the potential of learning dynamic aggregators by learning the parameters $\beta$, $p$, and even $y$ within GEN. Table 1 (d) *DyResGEN* reports the results of learning $\beta$, $\beta\&y$, $p$ and $p\&y$ for SoftMax_Agg, SoftMaxSum_Agg, PowerMean_Agg and PowerMeanSum_Agg respectively. In practice, $y$ is bounded from 0 to 1 by a sigmoid function. In all experiments, we initialize the values of $\beta$, $p$ to 1 and $y$ to 0.5 at the beginning of training. In order to show the improvement of the learning process, we also ablate experiments with fixed initial values. We denote aggregators with fixed initial values as *Fixed* and learned aggregators as *Learned*. We see that learning these variables consistently boosts the average performances of all the learned aggregators compared to the fixed initialized counterparts, which shows the effectiveness of learning adaptive aggregators. In particular, when $\beta$ is learned, DyResGEN-SoftMax achieves $86.15\%$ at 112 layers. We observe that DyResGEN-SoftMax outperforms the best ResGEN-SoftMax ($\beta = 10^4$) in terms of the average performance ($85.22\%$ *vs*. $85.17\%$). Interesting, we find generalizing the *sum* aggregation with PowerMean significantly improve the average performance from $84.56\%$ to $85.16\%$. We also put the best learned generalizing message aggregators in Table 1 (a) *ResGCN+* with gray color for a convenient comparison.

**Comparison with SOTA.** We apply our GCN models to six other OGB datasets and compare results with the published SOTA method posted on OGB Learderboard at the time of this submission (See Table 2). The methods include Deepwalk (Perozzi et al., 2014), GCN (Kipf & Welling, 2016), GraphSAGE (Hamilton et al., 2017), GIN (Xu et al., 2019b), GIN or GCN with virtual nodes, JKNet (Xu et al., 2019a), GaAN (Zhang et al., 2018), GatedGCN (Bresson & Laurent, 2018), GAT (Veličković et al., 2018), HIMP (Fey et al., 2020), GCNII (Ming Chen et al., 2020), DAGNN (Liu et al., 2020). The provided results on each dataset are obtained by averaging the results from 10 independent runs. It is clear that our proposed GCN models outperform SOTA on all four datasets. In two of these datasets (ogbn-proteins and ogbg-ppa), the improvement is substantial. The implementation details and more experimental results can be found in the Appendix.

Table 2: Comparisons with SOTA.* denotes that virtual nodes are used.

| | | | | | | | |
|---|---|---|---|---|---|---|---|
| | GraphSAGE | GCN | GaAN | | | | Ours |
| ogbn-proteins | $77.68 \pm 0.20$ | $72.51 \pm 0.35$ | $78.03 \pm 0.73$ | | | | $\mathbf{86.16 \pm 0.16}$ |
| | GraphSAGE | GCN | GaAN | GCNII | JKNet | DAGNN | |
| ogbn-arxiv | $71.49 \pm 0.27$ | $71.74 \pm 0.29$ | $71.97 \pm 0.24$ | $\mathbf{72.74 \pm 0.16}$ | $72.19 \pm 0.21$ | $72.09 \pm 0.25$ | $72.32 \pm 0.27$ |
| | GraphSAGE | GCN | ClusterGCN | GraphSAINT | GAT | | |
| ogbn-products | $78.29 \pm 0.16$ | $75.64 \pm 0.21$ | $78.97 \pm 0.33$ | $80.27 \pm 0.26$ | $79.45 \pm 0.59$ | | $\mathbf{81.64 \pm 0.30}$ |
| | GIN | GCN | GIN* | GCN* | HIMP | | |
| ogbg-molhiv | $75.58 \pm 1.40$ | $76.06 \pm 0.97$ | $77.07 \pm 1.49$ | $75.99 \pm 1.19$ | $78.80 \pm 0.82$ | | $\mathbf{78.87 \pm 1.24}$ |
| ogbg-molpcba | $22.66 \pm 0.28$ | $20.20 \pm 0.24$ | $27.03 \pm 0.23$ | $24.24 \pm 0.34$ | | | $\mathbf{27.81 \pm 0.38}$* |
| ogbg-ppa | $68.92 \pm 1.00$ | $68.39 \pm 0.84$ | $70.37 \pm 1.07$ | $68.57 \pm 0.61$ | $77.12 \pm 0.71$ | | $\mathbf{77.12 \pm 0.71}$ |
| | GraphSAGE | GCN | DeepWalk | | | | |
| ogbl-collab | $48.10 \pm 0.81$ | $44.75 \pm 1.07$ | $50.37 \pm 0.34$ | | | | $\mathbf{52.73 \pm 0.47}$ |

## 6 CONCLUSION

In this work, we proposed a differentiable generalized message aggregation function, which defines a family of permutation invariant functions. We identify the choice of aggregation functions is crucial to the performance of deep GCNs. Experiments show that existence of better generalized aggregators beyond *mean*, *max* and *sum*. Empirically, we show the effectiveness of training our proposed deep GEN models, whereby we set a new SOTA on several datasets of the challenging Open Graph Benchmark. We believe the definition of such a generalized aggregation function provides a new view to the design of aggregation functions in GCNs.

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

## A    DISCUSSION ON NETWORK DEPTH

**Depth & Residual connections.** Experiments in Figure 2 show that residual connections signif-icantly improve the training dynamic of deep GCN models. PlainGCN without skip connections suffers from vanishing gradient and does not gain any improvement from increasing depth. More prominent gains can be observed in ResGCN+ compared to ResGCN as models go deeper. No-tably, ResGCN+ reaches smallest training loss with 112 layers.This validates the effectiveness of pre-activation residual connections.

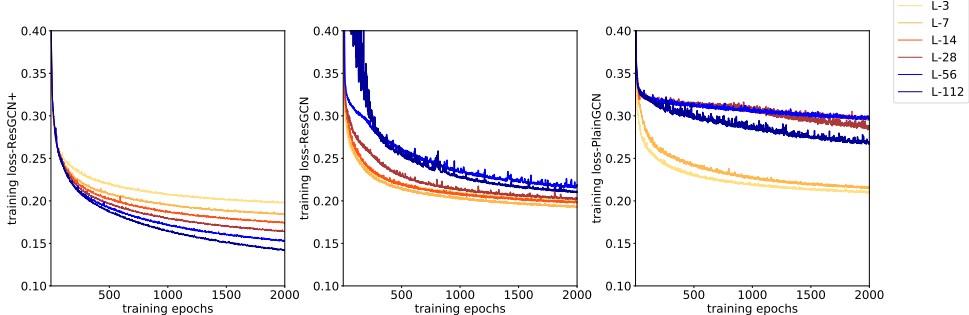

Figure 2: Training loss of PlainGCN, ResGCN and ResGCN+

**Depth & Normalization.** In our experiments, we find normalization techniques play a crucial role in training deep GCNs. Without normalization, the training of deep network may suffer from vanishing gradient or exploding gradient problem. We apply normalization methods such as BatchNorm (Ioffe & Szegedy, 2015) or LayerNorm (Ba et al., 2016) to normalize vertex features. In addition to this, we also propose a *message normalization* (MsgNorm) layer to normalize features on the message level, which can significantly boost the performance of networks with under-performing aggregation functions. The main idea of *MsgNorm* is to normalize the features of the aggregated message $\mathbf{m}_v^{(l)} \in \mathbb{R}^D$ by combining them with other features during the vertex update phase. Suppose we apply the MsgNorm to a simple vertex update function $\mathrm{MLP}(\mathbf{h}_v^{(l)} + \mathbf{m}_v^{(l)})$. The vertex update function becomes as follows:

$$\mathbf{h}_v^{(l+1)} = \boldsymbol{\phi}^{(l)}(\mathbf{h}_v^{(l)}, \mathbf{m}_v^{(l)}) = \mathrm{MLP}(\mathbf{h}_v^{(l)} + s \cdot \|\mathbf{h}_v^{(l)}\|_2 \cdot \frac{\mathbf{m}_v^{(l)}}{\|\mathbf{m}_v^{(l)}\|_2}) \qquad (5)$$

where $\mathrm{MLP}(\cdot)$ is a multi-layer perceptron and $s$ is a learnable scaling factor. The aggregated message $\mathbf{m}_v^{(l)}$ is first normalized by its $\ell_2$ norm and then scaled by the $\ell_2$ norm of $\mathbf{h}_v^{(l)}$ by a factor of $s$. In practice, we set the scaling factor $s$ to be a learnable scalar with an initialized value of 1. Note that when $s = \|\mathbf{m}_v^{(l)}\|_2 / \|\mathbf{h}_v^{(l)}\|_2$, the vertex update function reduces to the original form. In our experiment, we find *MsgNorm* boosts performance of under-performing aggregation functions such as *mean* and *PowerMean* on ogbn-proteins more than $1\%$. However, we do not see any significant gain on well-performing aggregation functions such as *SoftMax*, *SoftMaxSum* and *PowerMeanSum*. We leave this for our future investigation.

**Depth & Width.** In order to gain a larger representational capacity, we can either increase depth or width of networks. In this work, we focus on the depth instead of the width since it is more challeng-ing to train a deeper graph neural network compared to a wider one because of vanishing gradient (Li et al., 2019b) and over-smoothing (Li et al., 2018) problems. Deeper neural networks can learn to extract higher-level features. However, given a certain budget of parameters and computation, a well-designed wider networks can be more accurate and efficient than a deep networks. The trade-off of depth and width have already studied in CNNs (Zagoruyko & Komodakis, 2016). We believe that it is also important to study the width of GCNs to reduce the computational overhead.

**Depth & Receptive Flied & Diameter.** There are lots of discussion on whether depth can help for graph neural networks. In our experiments, we find that graph neural networks can gain better

performance with proper skip connections, normalization and aggregation functions. A interesting discussion by Rossi et al. (2020) argues that the receptive field of graph neural networks with a few layers can cover the entire graph since most of graph data are 'small-world' graphs with small diameter. Depth may be harmful for graph neural networks. In our experiment, we observe a different phenomenon. For instance, ogbn-proteins dataset with a relatively small diameter as 9 can gain improvement with more than 100 layers. However, what is the optimal depth and for what certain kind of graphs depth help more are still mysteries.

## B    PROOF FOR PROPOSITION 4

*Proof.* Suppose we have $N = |\mathcal{N}(v)|$. We denote the message set as $\mathbf{M} = \{\mathbf{m}_1, ..., \mathbf{m}_N\}$, $\mathbf{m}_i \in \mathbb{R}^D$. We first show for any message set, $\text{SoftMax\_Agg}_\beta(\mathbf{M}) = \sum_{j=1}^{N} \frac{\exp(\beta \mathbf{m}_j)}{\sum_{i=1}^{N} \exp(\beta \mathbf{m}_i)} \cdot \mathbf{m}_j$ satisfies Definition 2. Let $\rho$ denotes a permutation on the message set $\mathbf{M}$. $\forall \beta \in \mathbb{R}$, for any $\rho \star \mathbf{M} = \{\mathbf{m}_{\rho(1)}, ..., \mathbf{m}_{\rho(N)}\}$, it is obvious that $\sum_{i=\rho(1)}^{\rho(N)} \exp(\beta \mathbf{m}_i) = \sum_{i=1}^{N} \exp(\beta \mathbf{m}_i)$ and $\sum_{j=\rho(1)}^{\rho(N)} \exp(\beta \mathbf{m}_j) \cdot \mathbf{m}_j = \sum_{j=1}^{N} \exp(\beta \mathbf{m}_j) \cdot \mathbf{m}_j$ since the Sum function is a permutation invariant function. Thus, we have $\text{SoftMax\_Agg}_\beta(\mathbf{M}) = \text{SoftMax\_Agg}_\beta(\rho \star \mathbf{M})$. $\text{SoftMax\_Agg}_\beta(\cdot)$ satisfies Definition 2.

We now prove $\text{SoftMax\_Agg}_\beta(\cdot)$ satisfies Definition 3, *i.e.* $\lim_{\beta \to 0} \text{SoftMax\_Agg}_\beta(\cdot) = \text{Mean}(\cdot)$ and $\lim_{\beta \to \infty} \text{SoftMax\_Agg}_\beta(\cdot) = \text{Max}(\cdot)$. For the $k$-th dimension, we have input message features as $\{m_1^{(k)}, ..., m_N^{(k)}\}$. $\lim_{\beta \to 0} \text{SoftMax\_Agg}_\beta(\{m_1^{(k)}, ..., m_N^{(k)}\}) = \sum_{j=1}^{N} \frac{\exp(\beta m_j^{(k)})}{\sum_{i=1}^{N} \exp(\beta m_i^{(k)})} \cdot m_j^{(k)} = \sum_{j=1}^{N} \frac{1}{N} \cdot m_j^{(k)} = \frac{1}{N} \sum_{j=1}^{N} \cdot m_j^{(k)} = \text{Mean}(\{m_1^{(k)}, ..., m_N^{(k)}\})$. Suppose we have $c$ elements that are equal to the maximum value $m^*$. When $\beta \to \infty$, we have:

$$\frac{\exp(\beta m_j^{(k)})}{\sum_{i=1}^{N} \exp(\beta m_i^{(k)})} = \frac{1}{\sum_{i=1}^{N} \exp(\beta(m_i^{(k)} - m_j^{(k)}))} = \begin{cases} 1/c & \text{for } m_j^{(k)} = m^* \\ 0 & \text{for } m_j^{(k)} < m^* \end{cases} \tag{6}$$

We obtain $\lim_{\beta \to \infty} \text{SoftMax\_Agg}_\beta(\{m_1^{(k)}, ..., m_N^{(k)}\}) = c \cdot \frac{1}{c} \cdot m^* = m^* = \text{Max}(\{m_1^{(k)}, ..., m_N^{(k)}\})$. It is obvious that the conclusions above generalize to all the dimensions. Therefore, $\text{SoftMax\_Agg}_\beta(\cdot)$ is a generalized mean-max aggregation function. $\square$

## C    PROOF FOR PROPOSITION 5

*Proof.* Suppose we have $N = |\mathcal{N}(v)|$. We denote the message set as $\mathbf{M} = \{\mathbf{m}_1, ..., \mathbf{m}_N\}$, $\mathbf{m}_i \in \mathbb{R}_+^D$. We have $\text{PowerMean\_Agg}_p(\mathbf{M}) = (\frac{1}{N} \sum_{i=1}^{N} \mathbf{m}_i^p)^{1/p}$, $p \neq 0$. Clearly, for any permutation $\rho \star \mathbf{M} = \{\mathbf{m}_{\rho(1)}, ..., \mathbf{m}_{\rho(N)}\}$, $\text{PowerMean\_Agg}_p(\rho \star \mathbf{M}) = \text{PowerMean\_Agg}_p(\mathbf{M})$. Hence, $\text{PowerMean\_Agg}_p(\cdot)$ satisfies Definition 2. Then we prove $\text{PowerMean\_Agg}_p(\cdot)$ satisfies Definition 3 *i.e.* $\text{PowerMean\_Agg}_{p=1}(\cdot) = \text{Mean}(\cdot)$ and $\lim_{p \to \infty} \text{PowerMean\_Agg}_p(\cdot) = \text{Max}(\cdot)$. For the $k$-th dimension, we have input message features as $\{m_1^{(k)}, ..., m_N^{(k)}\}$. $\text{PowerMean\_Agg}_{p=1}(\{m_1^{(k)}, ..., m_N^{(k)}\}) = \frac{1}{N} \sum_{i=1}^{N} \cdot m_i^{(k)} = \text{Mean}(\{m_1^{(k)}, ..., m_N^{(k)}\})$. Assume we have $c$ elements that are equal to the maximum value $m^*$. When $p \to \infty$, we have:

$$\lim_{p \to \infty} \text{PowerMean\_Agg}_p(\{m_1^{(k)}, ..., m_N^{(k)}\}) = (\frac{1}{N} \sum_{i=1}^{N} (m_i^{(k)})^p)^{1/p} = (\frac{1}{N}(m^*)^p \sum_{i=1}^{N} (\frac{m_i^{(k)}}{m^*})^p)^{1/p} \tag{7}$$

$$= (\frac{c}{N}(m^*)^p)^{1/p} \xrightarrow{m^*>0} m^* \tag{8}$$

We have $\lim_{p \to \infty} \text{PowerMean\_Agg}_p(\{m_1^{(k)}, ..., m_N^{(k)}\}) = m^* = \text{Max}(\{m_1^{(k)}, ..., m_N^{(k)}\})$. The conclusions above hold for all the dimensions. Thus, $\text{PowerMean\_Agg}_p(\cdot)$ is a generalized mean-max aggregation function. $\square$

# D   ANALYSIS OF DYRESGEN

We provide more analysis and some interesting findings of DyResGEN in this section. The experimental results of DyResGEN in this section are obtained on ogbn-proteins dataset. We visualize the learning dynamic of learnable parameters $\beta$, $p$ and $s$ of 7-layer DyResGEN with SoftMaxSum_Agg$_{(\beta,y)}(\cdot)$ aggregator and PowerMeanSum_Agg$_{(p,y)}(\cdot)$ aggregator respectively. Learnable parameters $\beta$ and $p$ are initialized as 1 and $y$ are initialized as 0.5. Dropout with a rate of 0.1 is used for each layer to prevent over-fitting. The learning curves of learnable parameters of SoftMaxSum_Agg$_{(\beta,y)}(\cdot)$ are shown in Figure 3. We observe that both $\beta$ and $y$ change dynamically during the training. The $\beta$ and $y$ parameters of some layers tend to be stable after 1000 training epochs. Exceptionally, the 1-st layer learns a $\beta$ increasingly from 1 to 3.3 which learns a smaller $y \approx 0.1$ which make SoftMaxSum_Agg$_{(\beta,y)}(\cdot)$ behave more like a Max aggregation at the 1-th layer. PowerMean_Agg$_p(\cdot)$ aggregator also demonstrates a similar phenomena on learning $y$ in Figure 4. The learned $y$ of the 1-st layer and the last layer trends to be smaller than the initial value.

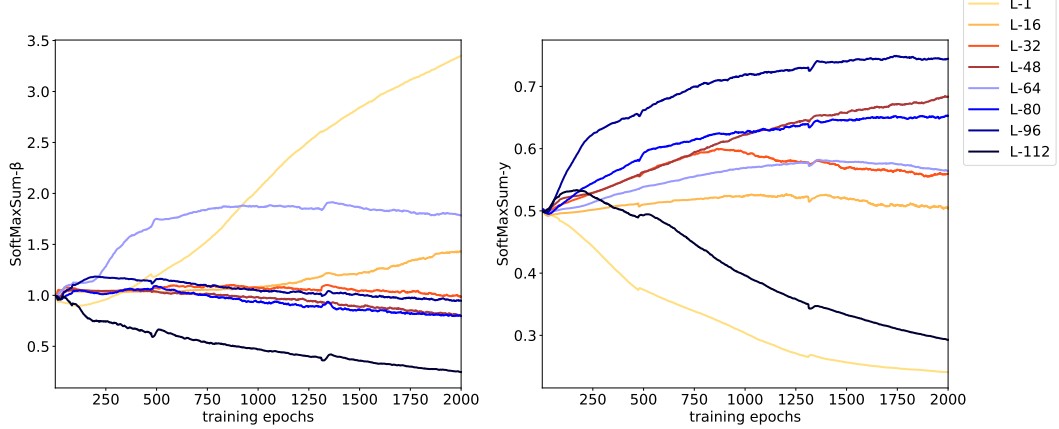

Figure 3: Learning curves of 112-layer DyResGEN with SoftMaxSum_Agg$_\beta(\cdot)$.

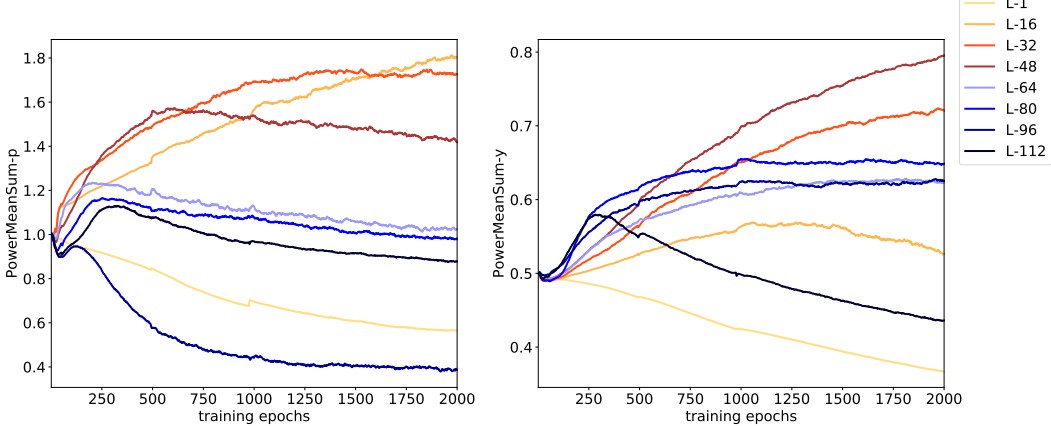

Figure 4: Learning curves of 112-layer DyResGEN with PowerMeanSum_Agg$_p(\cdot)$.

# E   MORE DETAILS ON THE EXPERIMENTS

In this section, we provide more experimental details on the OGB datasets (ogbn-proteins, ogbn-arxiv, ogbn-products, ogbg-molhiv, ogbg-molpcba, ogbg-ppa and ogbl-collab).

### E.1 DETAILS OF DATASETS

**Node Property Prediction.** Three chosen datasets are dealing with protein-protein association networks (ogbn-proteins), paper citation networks (ogbn-arxiv) and co-purchasing network (ogbn-products). Ogbn-proteins is an undirected, weighted, and typed (according to species) graph containing $132,534$ nodes and $39,561,252$ edges. All edges come with 8-dimensional features and each node has an 8-dimensional one-hot feature indicating which species the corresponding protein comes from. Ogbn-arxiv consists of $169,343$ nodes and $1,166,243$ directed edges. Each node is an arxiv paper represented by a 128-dimensional features and each directed edge indicates the citation direction. As an Amazon products co-purchasing network, ogbn-products is an undirected and unweighted graph which is formed by $2,449,029$ nodes and $61,859,140$ edges where nodes are products sold in Amazon that are represented by 100-dimensional features, and edges indicate that the connected nodes are co-purchased. For ogbn-proteins, the prediction task is multi-label and ROC-AUC is used as the evaluation metric. For ogbn-arxiv and ogbn-products, their prediction tasks are both multi-class and evaluated by accuracy.

**Graph Property Prediction.** Here, we consider three datasets, two of which deals with molecular graphs (ogbg-molhiv and ogbg-molpcba) and the other is biological subgraphs (ogbg-ppa). Ogbg-molhiv has $41,127$ subgraphs and ogbg-molpcba is much bigger which contains $437,929$ subgraphs. For ogbg-ppa, it consists of $158,100$ subgraphs and each subgraph is much denser in comparison to the other two datasets. The tasks of ogbg-molhiv and ogbg-molpcba are both binary classification while the prediction task of ogbg-ppa is multi-class classification. The former two are evaluated by the ROC-AUC and Average Precision (AP) metric separately. Accuracy is used to assess ogbg-ppa.

**Link Property Prediction.** We select ogbl-collab, an author collaboration network consisting of $235,868$ nodes and $1,285,465$ edges for link prediction task. Each node in the graph comes with a 128-dimensional feature vector representing an author and edges indicate the collaboration between authors. The task is to predict the future author collaboration relationships given the past collaborations. Each true collaboration is ranked among a set of $100,000$ randomly-sampled negative collaborations, and the ratio of positive edges that are ranked at $K$-place or above (*Hits@k*, $k$ is 50 here) is counted as the evaluation metric.

### E.2 DETAILS OF RESULTS AND IMPLEMENTATION

For a fair comparison with SOTA methods, we provide results on each dataset by averaging the results from 10 independent runs. We provide the details of the model configuration on each dataset. All models are implemented based on PyTorch Geometric (Fey & Lenssen, 2019) and all experiments are performed on a single NVIDIA V100 32GB.

**ogbn-proteins.** For both ogbn-proteins and ogbg-ppa, there is no node feature provided. We initialize the features of nodes through aggregating the features of their connected edges by a Sum aggregation, *i.e.* $\mathbf{x_i} = \sum_{j \in \mathcal{N}(i)} \mathbf{e}_{i,j}$, where $\mathbf{x_i}$ denotes the initialized node features and $\mathbf{e}_{i,j}$ denotes the input edge features. We train a 112-layer DyResGEN with $\text{SoftMax\_Agg}_\beta(\cdot)$ aggregator. A hidden channel size of 64 is used. A layer normalization and a dropout with a rate of 0.1 are used for each layer. We train the model for 2000 epochs with an Adam optimizer with a learning rate of 0.001.

**ogbn-arxiv.** We train a 28-layer ResGEN model with $\text{SoftMax\_Agg}_\beta(\cdot)$ aggregator where $\beta$ is fixed as 0.1. We convert this directed graph into undirected and add self-loop. Full batch training and test are applied. A batch normalization is used for each layer. The hidden channel size is 128. We apply a dropout with a rate of 0.5 for each layer. An Adam optimizer with a learning rate of 0.001 is used to train the model for 2000 epochs.

**ogbn-products.** A 14-layer ResGEN model with $\text{SoftMax\_Agg}_\beta(\cdot)$ aggregator where $\beta$ is fixed as 0.1 is trained for ogbn-products with self-loop added. We apply mini-batch training scenario by randomly partitioning the graph into 10 subgraphs and do full-batch test. For each layer, a batch normalization is used. The hidden channel size is 128. We apply a dropout with a rate of 0.5 for each layer. An Adam optimizer with a learning rate of 0.001 is used to train the model for 1000 epochs.

**ogbg-molhiv.** We train a 7-layer DyResGEN model with SoftMax_Agg$_\beta(\cdot)$ aggregator where $\beta$ is learnable. A batch normalization is used for each layer. We set the hidden channel size as 256. A dropout with a rate of 0.2 is used for each layer. An Adam optimizer with a learning rate of 0.0001 are used to train the model for 300 epochs.

**ogbg-molpcba.** A 14-layer ResGEN model with SoftMax_Agg$_\beta(\cdot)$ aggregator where $\beta$ is fixed as 0.1 is trained. In addition, the original model performs message passing over augmented graphs with virtual nodes added. A batch normalization is used for each layer. We set the hidden channel size as 256. A dropout with a rate of 0.5 is used for each layer. An Adam optimizer with a learning rate of 0.01 are used to train the model for 300 epochs.

**ogbg-ppa.** As mentioned, we initialize the node features via a Sum aggregation. We train a 28-layer ResGEN model with SoftMax_Agg$_\beta(\cdot)$ aggregator where $\beta$ is fixed as 0.01. We apply a layer normalization for each layer. The hidden channel size is set as 128. A dropout with a rate of 0.5 is used for each layer. We use an Adam optimizer with a learning rate of 0.01 to train the model for 200 epochs.

**ogbl-collab.** The whole model used to train on link prediction task consists of two parts: a 7-layer DyResGEN model with SoftMax_Agg$_\beta(\cdot)$ aggregator where $\beta$ is learnable and a 3-layer link predictor model. A batch normalization is used for each layer in DyResGEN model. We set the hidden channel size as 128. An Adam optimizer with a learning rate of 0.001 are used to train the model for 400 epochs.

## F  MORE FUTURE WORKS

We believe generalized aggregation functions will open a new view for designing aggregation functions in graph neural networks. Here we discuss some more potential directions as follows:

- Can we learn the parameters of generalized aggregation functions with a mete-learning method such as MAML (Finn et al., 2017)?

- What is the expressive power border of generalized mean-max-sum aggregation functions with respect to WeisfeilerLehman graph isomorphism test (Xu et al., 2019b)?

- Can we design Principal Neighbourhood Aggregation (PNA) (Corso et al., 2020) by combining multiple learnable aggregators from generalized aggregation functions?

