# OpenReview forum: "DeeperGCN: Training Deeper GCNs with Generalized Aggregation Functions"
_ICLR.cc/2021/Conference — Reject_

### Official Review · AnonReviewer2 · 2020-10-25
**Recommendation to weak accept**

**Rating:** 6
**Confidence:** 5

**Review:**

#####Summary#####

This work proposes a generalized aggregation function for graph neural networks. This generalized aggregation function can cover commonly used aggregation functions (i.e., mean, max, and sum) by particular setting of hyperparameters. Also, these hyperparameters can be learned with model in an end-to-end fashion instead of being predefined manually. The experimental results on OGB is impressive and can demonstrate the effectiveness of the proposed approaches.

#####Pros#####

(1) The proposed generalized aggregation functions are intuitively and empirically effective.

(2) The included experiments are comprehensive and can demonstrate the effectiveness of the proposed method.

(3) This paper is well organized and easy to follow.

#####Cons#####

(1) Although the provided experiments are comprehensive, some key experiments are missed. A. The proposed aggregation functions seem need more computations than simple mean/max/sum. A comparison of efficiency between proposed aggregation functions and existing simple functions should be included. B. The effectiveness of the whole model has been shown strongly. However, an ablation study to show the respective improvement brought by the increasing depth and proposed aggregation should be considered.

(2) It would be better to provide some theoretical support if possible.



#####Update######

Thank the authors for the response. I keep my score as 6.

---

> ### Author Response · Authors · 2020-11-24
> **Reply to Reviewer 2**
>
> Thanks for your helpful and insightful comments.
>
> 1, The proposed aggregation functions seem to need more computations than simple mean/max/sum. A comparison of efficiency between proposed aggregation functions and existing simple functions should be included.
>
> Thanks for your suggestion. We will add a discussion about the efficiency of proposed aggregation functions and conventional aggregation functions. In terms of the parameter-efficiency, both generalized Mean-Max aggregator and generalized Mean-Max-Sum aggregator only use 1 and 2 extra parameters respectively, which is negligible. However, the proposed generalized message aggregation functions need more computations. We measure the training time of a 7-layer model with different aggregation functions on ogbn-arxiv with full batch training for 2000 epochs. The results are shown as follows:
>
> | Aggregators         | Mean  | Max   | Sum   | PowerMeanSum |
> |---------------------|-------|-------|-------|--------------|
> | Test Acc (\%)       | 71.83 | 69.57 | 71.17 | 72.31        |
> | Traning Time (Mins) | 16.1  | 20.9  | 16.1  | 37.8         |
>
> PowerMeanSum outperforms all the other conventional aggregation functions. However, it consumes about 2.36X training time on ogbn-arxiv based on the current implementation.
>
> 2, The effectiveness of the whole model has been shown strongly. However, an ablation study to show the respective improvement brought by the increasing depth and proposed aggregation should be considered.
>
> (Note that the following reply between @@ is shared with the reply to Reviewer 3 and 4 partially)
>
> @@
>
> We can compare results horizontally and vertically in Table 1 (a) to see the effect of the increasing depth and the proposed aggregators respectively. We can see ResGCN+ coupling with mean aggregator resulting a poor performance on the ogbn-proteins dataset. The performance of ResGCN+(Mean) increases from 79.69\% to 83.40\% as the number of layers increases from 3 to 112. However, it can be observed that a very deep 112-layer ResGCN+(Mean) is outperformed by a 3-layer ResGCN+(SoftMax) on the ogbn-proteins (83.40\% v.s. 83.42\%). That is saying choosing the right aggregator is essential for training deep GNN architectures. With a learnable SoftMax aggregator, a 112-layer ResGCN+(SoftMax) outperforms a 112-layer ResGCN+(Mean) by a large margin (2.75\%). A similar phenomenon happens to ResGCN+(Max) on ogbn-arxiv. As the number of layers increases from 3 to 28, the performance of ResGCN+(Max) even degrades from 69.59\% to 66.91\%. In conclusion, increasing the depth blindly may not help with the improvement of the model's performance. The aggregators need to be carefully chosen on different datasets to unleash the power of deep GNNs. Generalized aggregation functions provide a larger function space and differentiability to overcome this issue.
>
> @@
>
> 3, It would be better to provide some theoretical support if possible.
>
> PNA [1] provides an interesting theoretical perspective to support the expressive power of generalized Mean-Max-Sum aggregation functions. By expressing the sum aggregator as the composition of a mean aggregator and a linear-degree amplifying scaler, the authors prove the mean aggregation composed with any scaling linear is an injective function on countable multisets. Therefore, generalized Mean-Max-Sum aggregation $|N(v)|^{y} \cdot \zeta_{x}(\cdot)$ is injective on countable multisets when $\zeta_{x}(\cdot)$ reduces to a Mean function and $y\neq0$. We would like to investigate more about the expressive power of generalized Mean-Max-Sum aggregation functions when $\zeta_{x}(\cdot)$ is not a Mean function in the future.
>
> [1] Corso, G., Cavalleri, L., Beaini, D., Liò, P. and Veličković, P. Principal neighbourhood aggregation for graph nets. NeurIPS 2020.

---

### Official Review · AnonReviewer4 · 2020-10-26
**Official Blind Review #4**

**Rating:** 4
**Confidence:** 3

**Review:**

The authors propose new and, in particular, parameterized aggregation functions for GNNs in order to especially support the construction of deeper GNNs. The paper is fairly understandable and the "deeper GNN" topic has gained more attention recently. However, in my opinion, the paper is missing the theoretical justification of its proposals and I have concerns about the results (details below). Hence, I do not vote for acceptance.

(+) The dynamic aggregation considered in Section 5.2 seems really interesting to me.

(-) Approach: The authors define several parameterized aggregation functions but the motivation is left unclear. Indeed, it seems that the main motivation for splitting the definitions in this way seems to be to generalize as many of the existing functions as possible. The paper is missing an explanation why these different aggregation functions are supposed to specifically support deeper GNNs.

(-) Experiments:
* The main table contains very many similar results but is missing standard deviation. This is especially strange given that the OGB code supports several folds.
* Table 2 is supposed to show the comparison with SOTA, but I claim that it is lacking. The initial OGB leaderboard contains only the most basic GNNs. However, when proposing an approach for creating deeper GNNs, the paper has to compare to this kind of models too. The row in which the authors compare to GCNII, JKNet, DAGNN, etc. actually seems to show that these models are as good/better.
----------------------------------------------

Smaller Comments:
- p.1 "The power of deep models become more evident with the introduction of more challenging and large-scale graph datasets" this claim should be supported by references. Why are they especially helpful for this kind of data?
- Table 1. The gray color is hardly readable and does not really allow for a "convenient comparison". I suggest to use bold face or anything else.

----------------------------------------------

Update after Rebuttal:
I have read the authors' response but do no change my scores.

The experimental results alone do not offer a theoretical justification. The paper does not have to contain the latter but then the evaluation would have to be exhaustive.

I see that the authors have done a huge number of experiments. However, if basic standards like standard deviation are neglected in order to run just more experiments, the results of the entire evaluation remain questionable. Similarly, related approaches have to be considered adequately to have an appropriate comparison to SOTA. The authors have not made them available yet.

---

> ### Author Response · Authors · 2020-11-24
> **Reply to Reviewer 4**
>
> Thank you for your thorough comments.
>
> 1, The motivation is unclear. The paper is missing an explanation why these different aggregation functions are supposed to specifically support deeper GNNs.
>
> (Note that the following reply between @@ is shared with the reply to Reviewer 3)
>
> @@
>
> The motivation for designing generalized aggregation functions is that the choice of aggregators may limit the performance of deep GCNs. In Table 1 (a), we can see ResGCN+ coupling with mean aggregator resulting a poor performance on the ogbn-proteins dataset. The performance of ResGCN+(Mean) increases from 79.69\% to 83.40\% as the number of layers increases from 3 to 112. However, we can observe that a very deep 112-layer ResGCN+(Mean) is outperformed by a 3-layer ResGCN+(SoftMax) on the ogbn-proteins (83.40\% v.s. 83.42\%). That is saying choosing the right aggregator is essential for training deep GNN architectures. With a learnable SoftMax aggregator, a 112-layer ResGCN+(SoftMax) outperforms a 112-layer ResGCN+(Mean) by a large margin (2.75\%). A similar phenomenon happens to ResGCN+(Max) on ogbn-arxiv. As the number of layers increases from 3 to 28, the performance of ResGCN+(Max) even degrades from 69.59\% to 66.91\%. Generalized aggregation functions provide a larger function space and differentiability to overcome this issue.
>
> Thanks for Reviewer 3 suggestion on the model-wise experiments. We further performed the experiments on three extra datasets including ogbg-molhiv, ogbg-ppa and ogbl-collab with a 7-layer ResGCN+ or a 7-layer ResGEN. The results are shown as follows:
>
> |             | ResGCN+    | |            |  ResGEN| |
> |-------------|---------|-------|-------|-------------------|-----------------|
> | Dataset     | Mean    | Max   | Sum   | SoftMax (Learned) | SoftMax (Tuned) |
> | ogbg-molhiv | 77.91   | 77.17 | 76.24 | 78.49             | 78.80           |
> | ogbg-ppa    | 74.09   | 74.32 | 75.42 | 75.18             | 75.96           |
> | ogbl-collab | 51.15   | 51.72 | 43.14 | 52.70             | 52.56           |
>
> All the results shown above are "the higher, the better". We study ResGCN+ with Mean, Max, and Sum aggregators and ResGEN with SoftMax aggregators. For the SoftMax aggregators, we include the learnable variant and the tuned temperature variant. For the tuned temperature t, we use t=0.1 on ogbg-molhiv, t=0.001 on ogbg-ppa and t=100 on ogbl-collab.
>
> The model-wise experiments further show the benefits of proposed aggregators. We can observe that conventional aggregators (Mean, Max, and Sum) perform very differently on different datasets. The performance of ResGEN with learnable SoftMax aggregators consistently achieves better or comparable results on different datasets. This shows the proposed aggregators can learn to adapt to different data.
>
> @@
>
> 2, The main table contains very many similar results but is missing standard deviation. This is especially strange given that the OGB code supports several folds.
>
> (Note that the following reply between @@ is shared with the reply to Reviewer 3)
>
> @@
>
> The provided results in Table 2 are obtained by averaging the results from 10 independent runs on each dataset. The mean and the standard deviation are given via following the guideline provided by the OGB leaderboards. We apologize that we could not conduct the same statistical analysis for all the ablation studies in Table 1 due to the limited computational resource. The ablation study includes 164 experiments totally on ogbn-proteins and each experiment takes more than 10 GPU hours on average. If we run all the experiments 10 times, it will take more than 16400 GPU hours, which is unrealistic. Although the results in Table 1 are obtained from one single run, we provide the average value over all the layers for the comparison between different aggregators. Moreover, the standard deviations on ogbn-proteins and ogbn-arxiv are usually relatively small (they are 0.16\% and 0.27\% in table 2). Based on the three-sigma rule, we believe the conclusions are reliable.
>
> @@
>
>
> 3, The initial OGB leaderboard contains only the most basic GNNs. However, when proposing an approach for creating deeper GNNs, the paper has to compare to this kind of models too. The row in which the authors compare to GCNII, JKNet, DAGNN, etc. actually seems to show that these models are as good/better.
>
> Thanks for your suggestions. We will add more comparisons with recent methods for training deeper GNNs in the next revision. We also would like to emphasize that our proposed aggregators can be applied to those mentioned GNNs architectures such as GCNII, JKNet, and DAGNN. In terms of the performance on ogbn-arxiv, our model is better than JKNet, DAGNN but worse than GCNII. However, GCNII uses 4 times more parameters compared to our model  (2,148,648(GCNII) v.s. 491,176(Ours)).
>
> 4, Smaller Comments
>
> Thanks for your valuable comments. We will adjust the text and the table accordingly in the next revision.

---

### Official Review · AnonReviewer1 · 2020-10-28
**Interesting direction but limited novelty of the proposed method.**

**Rating:** 4
**Confidence:** 5

**Review:**

This paper studies how to train deeper graph convolutional networks by using different aggregation function. Convolutional networks typically show better performance when getting deeper. However, this is not true in graph convolutional networks -- deeper graph neural networks usually show degenerate performance. So designing a deeper model for graph is both necessary and valuable.  This paper proposes a general aggregation function which summarizes sum, max, and softmax operations etc. I list the pros and cons as following.

Pros:
1. This paper studies an important problem for graph convolutional networks. Also, this paper proposes an interesting perspective to study this problem.
2. This paper conducts extensive experiments and analysis of different aggregation functions and their combinations.
3. The paper is well written.

Cons:
1. The greatest weakness of this paper is its novelty. Although studying aggregation function is interesting, this paper doesn't propose a new aggregation function beyond analyzing the combinations of existing ones. It would be great if authors can state the novelty of this paper compared to DeepGCNs. Also, what makes difference between the proposed aggregation function and softmax.
2. The proposed general aggregation function doesn't show significant improvement over softmax with learned temperature. Actually, the proposed powermean and softmaxsum are even worse than max.
3. The comparison to other methods (GraphSAGE and GCN) is not fair because they are shallow models while the model used in this paper is quite deep. I'd like to see more fair comparisons with the same model size. At least the authors should present the number of parameters and the size of their model.
4. This paper is based on DeepGCNs. DeepGCNs conducts experiments on point cloud recognition tasks and compare to DGCNN etc. I'd like to know why this paper doesn't consider point cloud recognition tasks as DeepGCNs tested on. Is it possible to compare to DGCNN/DeepGCNs on S3DIS?

---

> ### Author Response · Authors · 2020-11-24
> **Reply to Reviewer 1**
>
> Thank you very much for the constructive feedback.
>
> 1, "This paper doesn't propose a new aggregation function beyond analyzing the combinations of existing ones. It would be great if authors can state the novelty of this paper compared to DeepGCNs. Also, what makes the difference between the proposed aggregation function and softmax."
>
> We respectfully disagree this paper doesn't propose a new aggregation function. In this paper, we introduce a new class of differentiable aggregation functions beyond conventional Mean, Max, and Sum. We argue that the proposed generalized aggregation functions are not simple combinations of existing ones. The proposed generalized aggregation functions enlarge the function space (see Figure 1 for an illustration) and provide differentiability. Experiments also show that the existence of better generalized aggregators beyond Mean, Max, and Sum. More importantly, the definition of generalized aggregation functions provides a new perspective on the design of aggregation functions in GNNs.
>
> Compared to DeepGCNs, the main novelty in this paper is analyzing the effect of aggregation functions in training deep GNNs and proposing generalized aggregation functions to resolve the limitation of conventional aggregation functions. The importance of the selections of aggregation functions for training deep GNNs has not been shown in DeepGCNs. Besides, we also find applying the vanilla ResGCN to general graph datasets may still suffer from vanishing gradient problem. To this end, we adopt a pre-activation variant of residual connections for GCNs , which is referred to as ResGCN+. Please refer to Appendix A - Depth \& Residual connections for the detailed analysis.
>
> Regarding the difference between the proposed SoftMax aggregation function and softmax, the softmax function is known as a soft argmax. The proposed SoftMax-based aggregation function can be considered as a weighted sum function with weights from a soft argmax which has a differentiable temperature parameter.
>
> 2, "The proposed general aggregation function doesn't show significant improvement over softmax with learned temperature."
>
> To clarify, the SoftMax aggregation function is proposed in this paper as one of the four generalized aggregation functions. We do not claim all the generalized aggregation functions work equally well in practice. Empirically, we find the SoftMax and PowerMeanSum yield better results than PowerMean and SoftMaxSum. As mentioned in section 5.2, due to the numerical issues in PyTorch, PowerMean can not use large p. The reason is that large p may cause value overflow, which constrains the search space of PowerMean. Although PowerMean does not outperform Max, it still shows the merit of the learnable aggregators. In our experiments, the PowerMean aggregation is initialized as a Mean function with p=1. PowerMean clearly achieves better performance than the initialized Mean function (avg. 84.56\% (learned p)  v.s. 82.00\% (fixed p)).
>
> 3, "The comparison to other methods (GraphSAGE and GCN) is not fair because they are shallow models while the model used in this paper is quite deep. "
>
> Thanks for your great suggestion. Our original model indeed has a large model size in comparison to those shallow models. For a more fair comparison, we added an ablation study by reducing the number of parameters of our model.
>
> |      Model     |   Params  | Test ROC-AUC |
> |:--------------:|:---------:|:------------:|
> |       GCN      |   96,880  | 72.51 ± 0.35 |
> |    GraphSAGE   |  193,136  | 77.68 ± 0.20 |
> | Ours (Original) | 2,374,568 | 86.16 ± 0.16 |
> |  Ours (Small-1) |   80,446  | 83.50 ± 0.30 |
> |  Ours (Small-2) |   82,888  | 82.69 ± 0.21 |
>
> Our original model has 64 hidden channels and 112 layers and uses SoftMax aggregators. We created two compact models with fewer parameters than GCN by reducing the hidden sizes and the number of layers. Ours (Small-1) has 32 hidden channels and 14 layers. Ours (Small-2) has 16 hidden channels and 56 layers. Both of the compact models are smaller than GCN and GraphSAGE. The results above show our compact models still outperform GCN and GraphSAGE by a large margin (>5\%). This proves the efficiency of our proposed methods.
>
> 4, "Is it possible to compare to DGCNN/DeepGCNs on S3DIS? "
>
> We did not conduct experiments on S3DIS since our implementation was optimized for sparse graphs but not K-NN graphs. We agree that it is helpful to run experiments on point clouds. We performed experiments with a 7-layer ResDGCNN to show the effectiveness of proposed generalized aggregation functions. We compared a 7-layer ResDGCNN with Max aggregators and SoftMax aggregators on the area 5 of S3DIS. We observed ResDGCNN with SoftMax aggregators outperformed the counterpart slightly. We would investigate the effect on deeper models in the future.
>
> |      Model     | Test mIOU |
> |:--------------:|:---------:|
> | ResDGCNN(Max)  |   48.95   |
> | ResDGCNN(SoftMax) |   49.23   |

---

### Official Review · AnonReviewer3 · 2020-10-29
**Elegant solution but need more detailed experimental analysis**

**Rating:** 5
**Confidence:** 5

**Review:**

—Summary:

	The authors propose a generalized neighborhood message aggregation function for GNNs. The proposed choice of generalized aggregation functions is SoftMax and PowerMean, which generalizes Max and Mean functions and interpolates them. Additionally, they propose a variant of these two methods, which can also encompass the Sum function. By making the components of these generalized aggregator functions differentiable, the GNNs can choose an approximate instantiation of the aggregation function that best optimizes the task.
——

Pros:

	A straightforward and elegant solution
	The Paper is well written and is easy to follow.

——
Concerns:

	(i) It is not clear how generalized aggregation functions to aid training deeper GCNs
		Also, from the experiments, the baseline, ResGCN model’s performance also increases with layers and is falling short of the proposed model only by a small margin.

	(ii) Missing model wise results on OGB benchmark (Table 2)
		A detailed study is done only on one dataset, and in all other datasets, the best of the models alone is reported.

		For all the datasets, results for the following compared models from Table1 is required to understand the achieved performance improvement. Since the OGB benchmark is new and the reported GNN models on the OGB leaderboard were only run for 3 layers, it is crucial to analyze all the variants discussed here in detail to appreciate the performance gain achieved by the proposed generalized aggregation function.
 		- baselines: ResGCN and ResGCN+ with mean, max, sum,
 		- Fixed simple variant: Softmax, PowerMean, Softmax-sum, PowerMean-sum
		- Learnt variants: Softmax-sum and PowerMean-sum

	(iii) Benefits of the proposed model is inconclusive:
		Comparison with ResGCN+ is reported only for two datasets, and out of which, one dataset, the arXiv dataset, has results reported only for one of the aggregation functions. From the other protein dataset, the avg score for Max-ResGCN+ is 85.09, Learned PowerMean, and SoftMax based models scores are 84.56 and 85.22. There is both a drop and gain in performance with the adoption of the generalized aggregation functions. Hence, it is not clear whether both the generalized functions aid in improving performance without seeing the results on other datasets for all the four or two versions of the learned generalized aggregation functions (as in Table 1.d)

	(iv) Need a statistical significance test report.
	The results are very similar among models and are not clear whether there is any significant difference in choosing one over the other.

——
Questions during rebuttal:

	Kindly address the concerns raised above.
--- Post rebuttal:
I've read the author's response and there is no change in my scores.

- The given argument regarding better-generalized aggregation function aid in training deep GCNs is not clear and convincing.
- I agree that doing a detailed ablation study on a large dataset is expensive. In which case experiments on either synthetic or other smaller real-world datasets would be helpful.

---

> ### Author Response · Authors · 2020-11-24
> **Reply to Reviewer 3**
>
> We thank the reviewer for the insightful comments and concerns.
>
> 1, It is not clear how generalized aggregation functions to aid training deeper GCNs
>     Also, from the experiments, the baseline, ResGCN model’s performance also increases with layers and is falling short of the proposed model only by a small margin.
>
> (Note that the following reply between @@ is shared with the reply to Reviewer 4)
>
> @@
>
> The motivation for designing generalized aggregation functions is that the choice of aggregators may limit the performance of deep GCNs. In Table 1 (a), we can see ResGCN+ coupling with mean aggregator resulting a poor performance on the ogbn-proteins dataset. The performance of ResGCN+(Mean) increases from 79.69\% to 83.40\% as the number of layers increases from 3 to 112. However, we can observe that a very deep 112-layer ResGCN+(Mean) is outperformed by a 3-layer ResGCN+(SoftMax) on the ogbn-proteins (83.40\% v.s. 83.42\%). That is saying choosing the right aggregator is essential for training deep GNN architectures. With a learnable SoftMax aggregator, a 112-layer ResGCN+(SoftMax) outperforms a 112-layer ResGCN+(Mean) by a large margin (2.75\%). A similar phenomenon happens to ResGCN+(Max) on ogbn-arxiv. As the number of layers increases from 3 to 28, the performance of ResGCN+(Max) even degrades from 69.59\% to 66.91\%. Generalized aggregation functions provide a larger function space and differentiability to overcome this issue.
>
> @@
>
> 2, Missing model wise results on OGB benchmark.
>
> Thanks for suggesting the model-wise experiments. We performed the experiments on three extra datasets including ogbg-molhiv, ogbg-ppa, and ogbl-collab with a 7-layer ResGCN+ or a 7-layer ResGEN. The results are shown as follows:
>
> |             | ResGCN+    | |            |  ResGEN| |
> |-------------|---------|-------|-------|-------------------|-----------------|
> | Dataset     | Mean    | Max   | Sum   | SoftMax (Learned) | SoftMax (Tuned) |
> | ogbg-molhiv | 77.91   | 77.17 | 76.24 | 78.49             | 78.80           |
> | ogbg-ppa    | 74.09   | 74.32 | 75.42 | 75.18             | 75.96           |
> | ogbl-collab | 51.15   | 51.72 | 43.14 | 52.70             | 52.56           |
>
> All the results shown above are "the higher, the better". We study ResGCN+ with Mean, Max, and Sum aggregators and ResGEN with SoftMax aggregators. For the SoftMax aggregators, we include the learnable variant and the tuned temperature variant. For the tuned temperature t, we use t=0.1 on ogbg-molhiv, t=0.001 on ogbg-ppa and t=100 on ogbl-collab. Due to the limited time, we can not conduct all the experiments suggested. We will include them in the next revision. We hope the ablation study above is helpful.
>
> 3, Benefits of the proposed model are inconclusive.
>
> The model-wise experiments further show the benefits of proposed aggregators. We can observe that conventional aggregators (Mean, Max, and Sum) perform very differently on different datasets. The performance of ResGEN with learnable SoftMax aggregators consistently achieves better or comparable results on different datasets. This shows that the proposed aggregators can learn to adapt to different data.
>
> 4, Need a statistical significance test report.
>
> (Note that the following reply between @@ is shared with the reply to Reviewer 4)
>
> @@
>
> The provided results in Table 2 are obtained by averaging the results from 10 independent runs on each dataset. The mean and the standard deviation are given via following the guideline provided by the OGB leaderboards. We apologize that we could not conduct the same statistical analysis for all the ablation studies in Table 1 due to the limited computational resource. The ablation study includes 164 experiments totally on ogbn-proteins and each experiment takes more than 10 GPU hours on average. If we run all the experiments 10 times, it will take more than 16400 GPU hours, which is unrealistic. Although the results in Table 1 are obtained from one single run, we provide the average value over all the layers for the comparison between different aggregators. Moreover, the standard deviations on ogbn-proteins and ogbn-arxiv are usually relatively small (they are 0.16\% and 0.27\% in table 2). Based on the three-sigma rule, we believe the conclusions are reliable.
>
> @@

---

### Decision · Program_Chairs · 2021-01-07
**Final Decision**

**Decision:**

Reject

**Comment:**

One referee supports acceptance, whereas three referees lean towards rejection. All referees agree that the idea introduced in the paper is interesting but find that the motivation and evaluation of the proposed aggregation functions could be significantly strengthened. The rebuttal addresses R1's concerns about novelty and unfair comparisons, R2's concerns about computational efficiency of the methods, R3's concerns about motivation of the proposed approach and some missing baselines, and R4's concerns about motivation. However, the rebuttal does not address the reviewers' concerns related to improvements achieved by the proposed approach, statistical significance nor appropriate comparison with SOTA. I agree with the reviewers that the paper tries to address a relevant problem and proposes interesting ideas, which are worth exploring. However, after discussion, the referees agree that further work should be devoted to strengthen the contribution. I agree with their assessment and hence must reject. In particular, I would strongly recommend to follow their suggestions to either provide strong theoretical motivation to support the claims of the paper or work on a strengthened empirical evaluation, following OGB guidelines to report the std of the results and including a proper comparison with the state of the art.